# Identification and Validation of Novel Reference Genes in Acute Lymphoblastic Leukemia for Droplet Digital PCR

**DOI:** 10.3390/genes10050376

**Published:** 2019-05-17

**Authors:** Vanessa Villegas-Ruíz, Karina Olmos-Valdez, Kattia Alejandra Castro-López, Victoria Estefanía Saucedo-Tepanecatl, Josselen Carina Ramírez-Chiquito, Eleazar Israel Pérez-López, Isabel Medina-Vera, Sergio Juárez-Méndez

**Affiliations:** 1Experimental Oncology Laboratory, Research Department, National Institute of Pediatrics, Mexico City 04530, Mexico; kary_230794@hotmail.com (K.O.-V.); kattia.castrolpz@gmail.com (K.A.C.-L.); saucedo9116@hotmail.com (V.E.S.-T.); carina-rc@outlook.com (J.C.R.-C.); eleazarisraelp@yahoo.com.mx (E.I.P.-L.); 2Research Methodology Department, National Institute of Pediatrics, Mexico City 04530, Mexico; isabelj.medinav@gmail.com

**Keywords:** references genes, droplet digital PCR, gene expression, leukemia

## Abstract

Droplet digital PCR is the most robust method for absolute nucleic acid quantification. However, RNA is a very versatile molecule and its abundance is tissue-dependent. RNA quantification is dependent on a reference control to estimate the abundance. Additionally, in cancer, many cellular processes are deregulated which consequently affects the gene expression profiles. In this work, we performed microarray data mining of different childhood cancers and healthy controls. We selected four genes that showed no gene expression variations (*PSMB6*, *PGGT1B*, *UBQLN2* and *UQCR2*) and four classical reference genes (*ACTB*, *GAPDH*, *RPL4* and RPS18). Gene expression was validated in 40 acute lymphoblastic leukemia samples by means of droplet digital PCR. We observed that *PSMB6*, *PGGT1B*, *UBQLN2* and *UQCR2* were expressed ~100 times less than *ACTB*, *GAPDH*, *RPL4* and *RPS18*. However, we observed excellent correlations among the new reference genes (*p* < 0.0001). We propose that *PSMB6*, *PGGT1B*, *UBQLN2* and *UQCR2* are housekeeping genes with low expression in childhood cancer.

## 1. Introduction

Cancer is one of the most common health problems in the world and the National Cancer Institute (NCI) estimates 1,735,350 new cases in the United States in 2018; of these, 10,270 cases are childhood cancer (https://www.cancer.gov). Thus, the identification and implementation of new markers that help to diagnose, prognose and target cancer are urgent priorities. Acute lymphoblastic leukemia (ALL) is the most common childhood malignancy worldwide and in Mexico [1]. The five-year survival is > 90% in developed countries [2,3,4]; however, in developing countries such as Mexico, the survival rates are very low [5], possibly due to late diagnosis and several other factors, including risk classification, cytogenetics and immunological and molecular alterations [6,7,8,9].

In the postgenomics era, several groups have focused on omics to study different alterations in cancer. Currently, several molecular markers have been identified in a wide variety of malignancies, including SNPs, DNA gain and loss, epigenetic modification, coding RNA, noncoding RNA and protein expression and their modifications. The cytogenetic alterations in several types of cancer have an important impact on the clinical prognosis of ALL, such as *BCR-ABL* [10], *FUS-ERG* [11], *ETV6-RUNX1* [12], *E2A-PBX1* [13] and *KMT2A-AFF1*. However, less than 20% of leukemia patients have an alteration.

Gene expression (GE) is a spectacular cell process that is tissue-driven and the dynamics of GE are modulated by several factors that include: environmental, microenvironmental, intracellular or extracellular processes, among others. Nevertheless, RNA diversity is not completely clear; bioinformatic studies have shown that only ~2% of the transcriptome promotes protein diversity, and the alternative splicing (AS) of mRNA provides an exceptional capacity to generate a complex proteome. Additionally, RNA expression has been used as a molecular marker in diverse malignancies, such as mendelian disorders [14], tuberculosis [15] and some types of cancer [16]. Nevertheless, its measurement is not completely uniform because it is dependent on cellular conditions, such as stress, temperature, O_2_ concentrations and overgrowth in vitro, which could influence modifications in GE. Thus, it is very important to establish methods for RNA quantification and reference genes (calibrator or housekeeping) that do not show variations between patients and healthy conditions. The expression level of specific transcripts in clinical diagnosis plays an important role in prognosis, treatment assignment and overall survival.

Furthermore, the GE level could be influenced by several technical factors, including contamination by alcohol, phenol, proteins; RNA integrity; and cDNA synthesis. In addition, RNA quantification could have some technical variations caused by RNA absolute quantification. Thus, the use of reference genes (housekeeping) is very important for normalizing GE and detecting changes in the expression of potential biomarkers. Third-generation PCR, known as droplet digital PCR (ddPCR), has proven to be a powerful tool for determining gene expression. It has higher accuracy, sensitivity and several advantages because it has minor inhibitory effects on PCR and internal normalization is unnecessary for detection, however it is recommendable for gene expression analysis with ddPCR [17]. It provides a great opportunity to absolutely measure the nucleic acid (DNA/RNA) and identify rare allelic variants, nonabundant RNA transcripts and copy number variations (CNVs), among others. This platform has proven to be a powerful tool with the ability to precisely count the absolute abundance of transcripts or DNA. The molecular abundance is critical in the diagnosis of several pathologies, and the clinical relevance in prognosis, such as detecting HER2 and EEF2 in breast cancer [18], improves the precision and accuracy of diagnosis.

Reports of potential molecular GE markers in ALL remain uncertain due to the lack of uniformity and reproducibility in the method criteria of real-time PCR, which focus mainly on normalization genes, and then the measured level expression of candidate genes in the study. The selection of a housekeeping gene is the first step in the normalization of an interest gene. However, several studies on RNA transcriptomics analysis in ALL have reported using a single and conventional constitutive gene; unfortunately, it is still common practice today [19,20]. This failure can lead to inaccuracy and miscalculation on GE and poor reproducibility. Therefore, it is necessary to improve critical variables in the quantitative transcripts to improve the precision and accuracy of potential GE in ALL [21].

On the other hand, ddPCR is a powerful method for absolute quantification of DNA/RNA. Moreover, GE is not constant in all transcriptomes; some transcripts are overexpressed and some others are poorly expressed. Thus, the aim of this work was to identify new candidate RNA transcripts with constitutive expression levels obtained by microarray data mining that did not have variations in the level expression using ddPCR in ALL. We optimized the ability to digitally count potential RNA transcripts and determined homogeneous expression versus level expression of housekeeping genes for ALL. These high quantitative and precise constitutive transcripts in ALL allow for the availability and validation of ALL samples using ddPCR. Therefore, it is of great importance to develop and identify new reference genes (NRGs) with low expression levels in childhood cancer, in special ALL and then have precise comparisons with classical reference genes (CRGs) of low or high expression using ddPCR.

## 2. Materials and methods

### 2.1. Gene Expression Data Mining

The data were obtained from the ArrayExpress and we selected the Affymetrix GeneChip 1.0. We downloaded 768 microarray experiments that corresponded to childhood malignancy, healthy tissue and cancer cell lines. The data included in the study are listed in (Appendix A). Then, we detected microarray quality control (MQC) variations according to a previous study [22,23,24]. We excluded all samples that showed changes in the intensity signal of the microarray controls, similar to the previous analysis [24]. Our QC analysis included 193 microarrays in the next analysis (Appendix A). The microarray data analysis was archived using Partek Genomics version 6.6, Santa Clara, CA, USA. Bioinformatic analysis was performed using quantile normalization, probeset summarization by Median Polish, background correction by robust microarrays analysis (RMA) and, finally, the data underwent log_2_ transformation. To identify genes that do not vary in expression levels by comparison between healthy tissues and tumors (including cancer cell lines), we used the exon intensity signals analyzed by ASANOVA with an unadjusted *p*-value > 0.95.

### 2.2. Tumor Samples

In this study, 40 samples of bone barrow diagnosed with ALL were obtained with previous signed informed consent. The protocol was approved by the Institutional Ethics Committee (INP protocol 060/2016) in accordance with the Declaration of Helsinki. The bone marrow samples were treated with lymphoprep density gradient medium (STEMCELL Technologies, USA) according to the protocol for cell pellets after RNA isolation. We used RNA from the SUP-B15 cell line to standardize the methodologies in this study. SUP-B15 was cultured in Iscove’s modified Dulbecco’s medium according to the manufacturer’s instructions (ATCC). The culture medium was supplemented with 20% fetal bovine serum (Biowest, Riverside, Kansas, USA) and penicillin/streptomycin (100 U/mL, ATCC, Manassas, VA, USA). Cells were cultured at 37 °C, 5% CO2 and a humidified atmosphere.

### 2.3. RNA Extraction

The cultured cells and patient cells were disrupted using a TissueLyser system (Qiagen, Valencia, CA, USA) for 60 s at 25 Hz. Total RNA was extracted using 1 mL of TRIzol reagent (Thermo Fisher Scientific, Waltham, MA, USA) according to the manufacturer’s instructions. RNA was resuspended in 45 μL of DPEC water and quantified using a NanoDrop One UV-Vis Spectrophotometer (Thermo Fisher Scientific, Waltham, MA, USA).

Then, RNA was quantified with the Qubit RNA HS Assay Kit (Thermo Fisher Scientific, Waltham, MA, USA) using a Qubit Fluorometer (Thermo Fisher Scientific, Waltham, MA, USA) according to the protocol. RNA samples were stored at −70 °C until further use.

### 2.4. cDNA Synthesis

Before cDNA synthesis, RNA samples were subjected to DNase treatment (Thermo Fisher Scientific, Waltham, MA, USA), which included 1000 ng of total RNA, 1 μL of 10× Buffer and 1U of DNase. DEPC-treated water was added for a final reaction volume of 10 µL. The reaction was incubated at 37 °C for 30 min, the cycle was stopped, 1 μL of 50 mM EDTA was added and the reaction was incubated at 65 °C for 10 min. cDNA synthesis was performed using the RevertAid Transcriptase Kit (Thermo Fisher Scientific, Waltham, MA, USA). The master mix contained 1× reaction buffer, 1 mM dNTP Mix, 100 pmol Random hexamers, 10 U RiboLock RNase Inhibitor and 200 U RevertAid Reverse Transcriptase and DEPC water up to 20 µL. The reactions were incubated at 25 °C for 10 min, 42 °C for 60 min and 70 °C for 10 min. The final concentration of cDNA samples was 50 ng/µL.

### 2.5. RT-PCR Amplification

We evaluated the cDNA synthesized using 25 ng of cDNA for end point PCR. The reaction contained 1× KAPA2G Mix, 0.5 μM forward and reverse primers and DEPC water up to 15 µL. The primers are listed in Table 1. The amplification program consisted of predenaturation at 95 °C for three minutes, followed by 38 cycles of 95 °C for 15 s, extension for 15 s (Tm for each constitutive transcript are shown in Table 1) and 72 °C for 15 s, with a final extension of 72 °C for five minutes in a Proflex PCR System thermal cycler (Applied Biosystem Inc., Foster City, CA, USA). The PCR products were separated by electrophoresis in 2.0% agarose gels (g/v) and stained with Sybr Gold (Thermo Fisher Scientific, Waltham, MA, USA) (1:10,000) at 90 volts for 35 min in 0.5% TAE solution. The amplification sequence was as follows: RPL4-a and RPS18-a first, followed by the rest of the genes listed with the letter q for quantitative RT-PCR.

### 2.6. Quantitative RT-PCR

Thereafter, cDNA of the SUP-B15 cell line was analyzed by quantitative RT-PCR through amplification of *ACTB*, *GAPDH*, *RPL4*, *RPS18*, *PGGT1B*, *PSMBG*, *UBQLN2* and *UQCR2* (Table 1). We used the Kapa Sybr Fast qPCR Master Mix 2× Kit (Kapa Biosystems Inc., Wilmington, MA, USA). The reaction included master mix with the addition of DEPC water up to 10 µL, 2× Kapa Sybr Fast reagent qPCR Master Mix containing MgCl_2_ at a final concentration of 2.5 mM, 10 μM forward and reverse primers and 10 ng of cDNA. Quantitative RT-PCR was performed on a Step One Real-Time PCR System (Applied Biosystems Inc., Foster City, CA, USA). The reactions were incubated at 95 °C for 10 min, then 40 cycles for amplification at 60 °C with a denaturation at 95 °C for 15 s and finally a melting curve at 53 °C for 1 min with a quantification every 0.3 °C for 15 s up to 95 °C.

### 2.7. Droplet Digital PCR

ddPCR was performed using QX200 ddPCR EvaGreen SuperMix (Bio-Rad, Hercules, CA, USA) in a QX200 Droplet Digital PCR System (Bio-Rad). Briefly, a PCR mixture containing 10 µL of 2× QX200 ddPCR EvaGreen SuperMix, 0.5 µL of each set of primers at 5 µM, 4 µL of cDNA and H_2_O necessary for a total reaction of 20 µL. The Automated Droplet Generator loaded 20 µL of each reaction mix in the DG8 Cartridge onto QX200 Droplet Generation system for droplet generation. The droplets were transferred into a 96-well plate and subsequently amplified using the following cycling parameters: predenaturization step at 95 °C, 35 amplification cycles of 95 °C for 15 s, 60 °C for 30 s, one cycle of 4 °C for 30 s, 90 °C for 5 min and finally a hold at 4 °C. Finally, the dropletized PCR in a 96-well plate was read by the Q×200 Droplet Reader. The Poisson-corrected determination of template concentration was calculated using QuantaSoft™ Analysis Pro Software (v1.0, Bio-Rad). For quantification, a minimum of 10,000 acceptable droplets per 20 µL reaction was used, followed by manual selection of positive and negative droplet populations. All experiments included a no-cDNA template control (NTC) and 0.2 ng/µL of cDNA for CRGs, and 5 ng/µL of cDNA was used to quantify the number of copies/20 µL by well for NRGs.

### 2.8. Statistical Analysis

Variables were assessed using the Kolmogorov–Smirnov Z test to examine sample distribution. We correlated the absolute quantifications of the GE among *RPL4*, *RPS18*, *ACTB*, *GAPDH*, *UQCR2*, *PSMB6*, *UBQLN2* and *PGGT1B*. The correlation was achieved using the Pearson correlation coefficient. We consider statistical significance *p* < 0.05. Data analysis was performed using SPSS v.25 software for Macintosh (IBM Corp., Armonk, NY, USA). Curve fitting was performed in GraphPad Prism by least squares optimization.

## 3. Results

### 3.1. New Candidates Reference Genes in Pediatric Cancer

First, we performed data mining of different pediatric tumors and control tissues using Affymetrix GeneChip 1.0. Our data mining identified and downloaded 768 microarrays, including osteosarcoma, Hodgkin lymphoma, rhabdomyosarcoma, retinoblastoma, neuroblastoma, medulloblastoma, leukemia, bone marrow, retina, fibroblasts, skeletal muscle, bone, liver and different cancer cell lines of these tumors (Appendix A). First, we identified the relationship by sample among microarray internal controls according to the manual of GeneChip 1.0 as follows: bioB, bioC, bioD and Cre at final concentrations of 1.5, 5, 25 and 100 pM, respectively. PolyA RNA controls were evaluated using Dap, Thr, Phe and Lys to final concentrations of 1:7500, 1:25,000, 1:50,000 and 1:100,000, respectively, as previously reported [24]. Our results show that ~75% of the available data showed differences in relation concentrations of almost one hybridization and/or polyA RNA control. Thus, we selected 193 microarray files that showed no differences among the controls (Appendix A).

Then, we classified the microarray files in healthy tissues (control, *n* = 45) and cancer tissues (tumor, *n* = 148) and we performed a comparative analysis using a control as a reference; however, we selected the transcripts without varying GE. Our analysis showed that 189 transcripts from healthy and tumor samples did not show significant changes in GE. After visual inspection, we selected 10 transcripts (Figure 1, Appendix A), however only four transcripts (*PSMB6*, *PGGT1B*, *UBQLN2* and *UQCRC2*, Figure 1) were validated by quantitative RT-PCR and droplet digital PCR.

### 3.2. ddPCR Assay Optimization for Analysis of New Reference Genes

To contrast our results, we selected four reported CRGs (*GAPDH*, *ACTB*, *RPS18* and *RPL4*). Thus, we compare the GE by end point PCR. Although all samples were quantified by spectrometry and fluorometry, we observed some variations in GE, most likely due to sample conditions and RNA quality, among others. Thus, we evaluated all transcripts using qRT-PCR using five serial dilutions with an initial concentration of 50 ng/uL, with a factor dilution of 1:5. The results showed good kinetic and linearity of amplification for all genes that were evaluated. We observed no unspecific amplicons in the melting curves for the positive samples and we observed apparent dimer primers in *RPS18*, *PSMB6P* and *PGGT1B* (Figure 2).

Next, we established ddPCR conditions for CRGs and NRGs as follows. First, we optimized the assay conditions to minimize droplet rain in terms of cDNA concentration, annealing temperature and primer concentration. We constructed an amplification curve using the SUP-B15 cell line for the eight transcripts using three samples tested with a serial dilution of 1:3. We obtained good linearity for eight genes with a regression coefficient of 0.99. Second, we plotted ng/reaction versus number of copies/well for eight reference genes and, interestingly, we observed that CRGs are expressed ~100 times more than NRGs (Figure 3). Additionally, we observed that the reaction was saturated to 12 ng/well for CRGs (Appendix A), while the NRGs showed low expression at the same template concentration (Appendix A), avoiding droplet rain in ddPCR for a highly concentrated sample.

### 3.3. Reference Genes Validation in Acute Lymphoblastic Leukemia

We quantified the absolute RNA expression of all evaluated transcripts. Our results showed that 0.8 ng/well is enough to determine the expression of these transcripts (CRGs), while 20 ng/well is enough to evaluate the transcripts with low expression (NRGs). After that, we evaluated the eight transcripts in 40 ALL samples. We observed a different copy number of reference transcripts, although all samples were quantified by spectrometry and fluorometry, of which eight representative samples of all reference genes were plotted (Figure 4). We observed differences in GE between CRGs and NRGs by sample. These results suggest that the ddPCR assay allowed us to discriminate genes with low and high expression.

### 3.4. Comparative Expression Between CRGs and NRGs in Leukemia

To confirm whether the expression of each type of gene was the same, we plotted a bar graph to show paired data for the same sample. The scatterplots shown in the y-axis represent the number of copies/ng normalized for CRGs and NRGs. First, each CRGs (*RPL4*, *RPS18*, *GAPDH*, *ACTB* and *GAPDH*) were compared with the four NRGs (*PSMB6*, *PGGT1B*, *UBQLN2* and *UQCR2*) in the copies/ng normalized in all samples that were evaluated. We systematically reviewed all comparisons and each scatterplot revealed the same patterns of change and clearly showed that the level of expression of CRGs was higher than that of NRGs (Figure 5). However, the level of expression for each CRGs in 40 ALL samples was proportional with respect to NRGs. This outcome revealed that patterns of expression level were similar between CRGs and NRGs. For this, it was necessary to evaluate the correlation of the absolute quantification of the expression between CRGs and NRGs (Figure 6). As expected, we obtained a high correlation between the gene studies, with the minimum value of 0.782 for *PSMB6* versus *ACTB*, while the highest correlation value was 0.987 for *RPL4* versus *PGGT1B*, both with *p*-values < 0.0001. The Spearman correlation helped to identify that all NRGs have a high value correlation with all CRGs; interestingly, *UQCR2* and *PGGT1B* showed a major correlation with all CRGs.

## 4. Discussion

Cancer is an important public health problem worldwide. In Mexico, leukemia is the most common childhood malignancy. There has been an increase in the number of new cases diagnosed, suggesting that the number of patients will be higher in the coming years. The challenge in cancer is the detection and sensitivity of the biomarkers. Many studies have focused on the identification of molecular markers in several types of cancer using microarray and next-generation sequencing.

In the last year, public data has been useful to identify molecular patterns in several types of cancer, mutations [25], AS [26] and molecular classification [24]. We performed microarray data mining of several types of childhood cancers and we inspected the quality control microarrays and observed that 70% of the data were rejected as we previously reported [22].

On the other hand, qRT-PCR quantification was performed based on a standard curve and the GE results are expressed as a fold change [27,28,29]. ddPCR is a robust methodology that has revolutionized nucleic acid quantification in providing sensitivity detection [30,31,32,33]. In ddPCR, each reaction is analyzed independently, the reaction is fractionated and the account positives may then use a Poisson correction to determine the absolute counts by target present in each sample [34] and facilitate measurement of individual target molecules [32]. Thus, ddPCR is a powerful method that achieves reproducibility, accuracy and sensitivity for the detection of molecular markers, disease monitoring and viral load, among others [32]. Therefore, ddPCR has been employed in the identification of low abundance molecules, such as fusion genes, minimal residual disease, SNPs and GE, among others [32,35,36].

RNA is a dynamic molecule of which its expression is tissue-dependent and the RNA quantification is dependent on cDNA synthesis efficiencies, sample degradation and RNA isolation [37]. Thus, it is very important to use the reference genes to normalize GE. Additionally, ddPCR provides high sensitivity for the detection and expression of relevant transcripts which is very important for normalization versus genes of equal or similar expression. Therefore, the comparison between genes of low and high expression would be incompletely comparable data because in ddPCR analysis, the determination of positive and negative droplets is a critical step; thus, the cDNA will be adjusted for unsaturation of the positive drop.

Several reports have focused on the comparison of the reference genes between qPCR versus ddPCR [17,38]. However, there are no reports about the use of the reference genes as normalizers of GE using ddPCR. In this study, we established the amount of cDNA/well in eight transcripts to be absolute quantification which allowed us to identify that the CRGs are expressed ~100 times more than NRGs (Figure 3). Previously, the transcript accounts were evaluated in the leukemia cell line SUP-B15 and subsequently normalized to 20 ng/well (Figure 3). Our results showed that *ACTB* was the most highly expressed gene, with the following relationships: 1:2.1, 1:6.4, 1:8.3, 1:73, 1:88, 1:263, 1:404, *ACTB*:*GAPDH*, *ACTB*:*RPL4*, *ACTB*:*RPS18*, *ACTB*:*UQCR2*, *ACTB*:*PSMB6*, *ACTB*:*UBQLN2* and *ACTB*:*PGGT1B*.

To date, there are a few reference genes to use in ddPCR for GE. However, the validation of GE in ALL showed that the expression level among samples was different despite the fact that the samples were quantified by fluorometry and spectrometry and the samples were adjusted to 1 µg for cDNA synthesis (Figure 4). These results suggest that some technical factors affect the efficiency of the RT reaction or are characteristic *per se* of the sample [37]. However, when we analyzed the expression of the eight transcripts by sample, we observed that the proportion of the expression is conserved (Figure 5). Some reports have shown that the reference genes change the GE between study conditions when a relative quantification is performed by qPCR [39].

In our study, we analyzed the expression of eight reference transcripts using qRT-PCR and ddPCR. After that, we validated the expression in 40 patients with ALL for ddPCR and the CRGs were expressed ~100 times more than the NRGs. Additionally, we observed a high correlation among the eight transcripts analyzed (Figure 6). Our results suggest that the new transcripts could be used as reference genes in ALL.

In this work, we propose four NRGs that could be used to normalize genes of which have expression levels that are inconspicuous and provide a better comparison between samples and experimental conditions. The RNA molecule is very dynamic and its quantification is dependent on cDNA synthesis, RNA integrity and characteristics of the sample. Therefore, the absolute RNA quantification of the relevant genes in cancer is necessary to normalize a reference gene with a similar level of expression to obtain accurate results.

## 5. Conclusions

We validated the expression of the eight reference genes in 40 ALL patients and found that the classical reference gene had a high level of expression in contrast to the four NRGs *PSMB6*, *PGGT1B*, *UBQLN2* and *UQCR2*, which showed a high correlation with CGRs. Finally, we propose these new genes as housekeeping genes in childhood cancer. Therefore, the use of these transcripts as a normalizer will be dependent on the target expression level.

## Figures and Tables

**Figure 1 genes-10-00376-f001:**
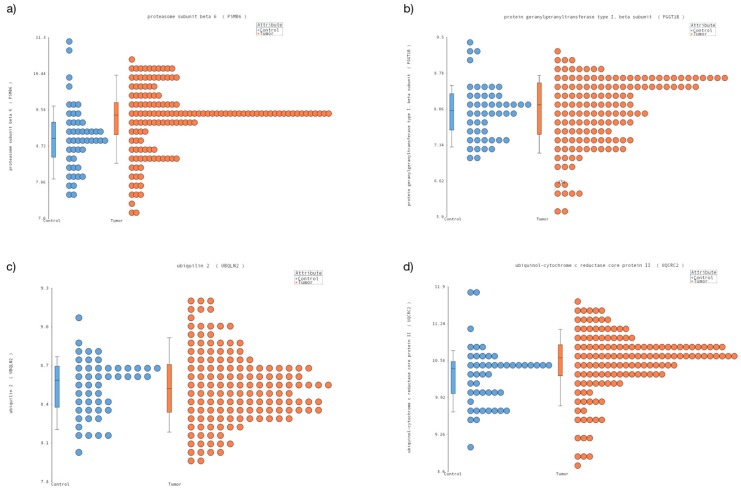
Nonvarying genes were expressed in childhood cancer and healthy tissue. The dot plot shows the level of expression on the *x* axis. (**a**) *PSMB6* (**b**) *PGGT1B*, (**c**) *UBQLN2*, (**d**) *UQCR2*. In the four transcripts, we observed homogenous expression between healthy tissues and cancer samples. The blue dots show the healthy tissue and the orange dots show the childhood cancer.

**Figure 2 genes-10-00376-f002:**
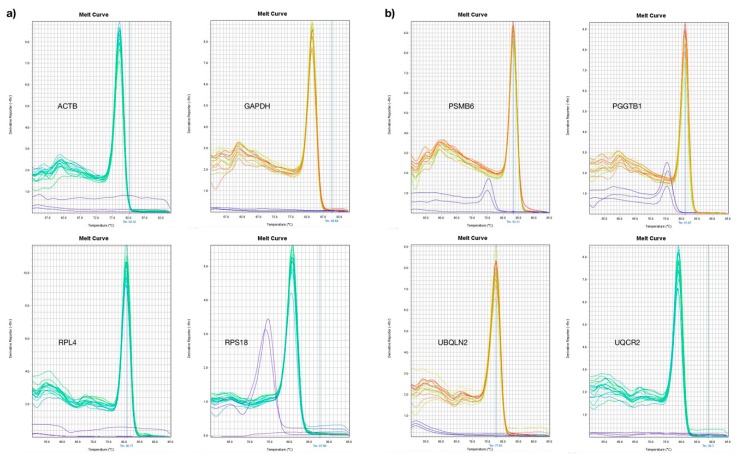
Melting curves of the expressed nonvarying genes. (**a**) The classical reference genes *ACTB*, *GAPDH*, *RPL4* and *RPS18* are shown in the figure. We used five serial dilutions of cDNA SUP-B15 and we observed one curve as expected. Only RPS18 showed up as a different curve in the no-template control, probably due to primer dimers. (**b**) The plot shows the new reference genes *PSMB6*, *PGGT1B*, *UBQLN2* and *UQCR2*. We observed one curve in the positive dilution. In *PSMB6* and *PGGT1B*, we observed other curves in the no-template control, probably caused by primer dimers.

**Figure 3 genes-10-00376-f003:**
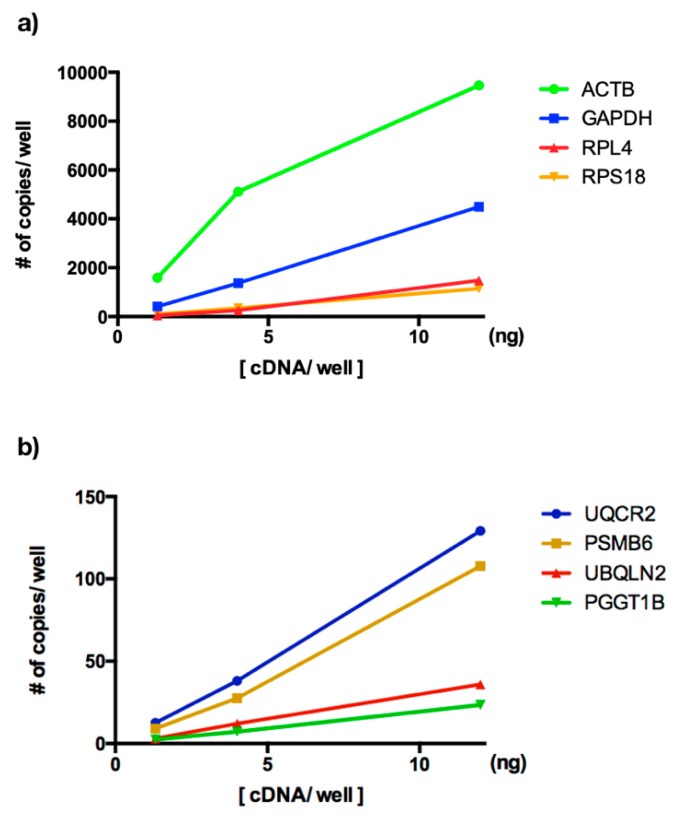
Amplification curves by absolute quantification. The *y*-axis shows # copies/well and the axis *x*-axis shows the cDNA concentration used in ddPCR. (**a**) Copy number transcript by well of *ACTB, GAPDH, RPL4* and *RPS18*. *ACTB* was the most highly expressed housekeeping gene. (**b**) Copy number transcript by well of *PSMB6*, *PGGT1B*, *UBQLN2* and *UQCR2*. *UQCR2* was the most highly expressed following *PSMB6*.

**Figure 4 genes-10-00376-f004:**
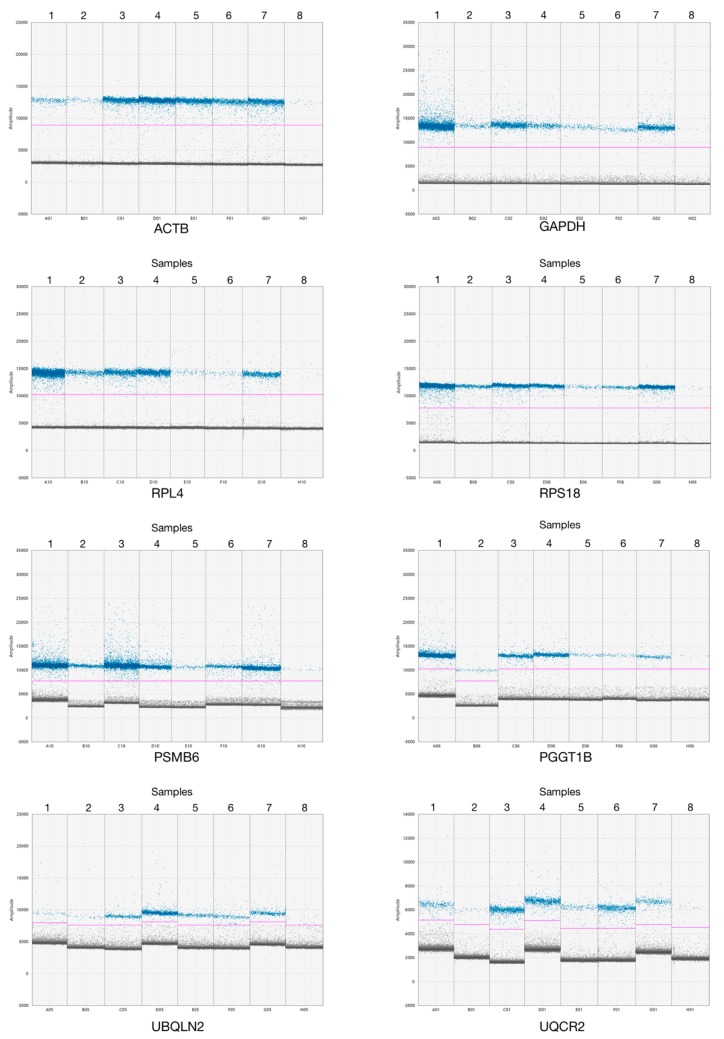
Droplet digital PCR amplification of the eight patients with acute lymphoblastic leukemia (ALL). Each plot represents the amplification of the eight reference genes evaluated in eight samples. The *y*-axis shows the amplitude signal and the *x*-axis shows the samples. Blue dots indicate positive amplification droplets and gray dots indicate negative amplification droplets. The pink line represents the cut-off of the positive and negative droplets.

**Figure 5 genes-10-00376-f005:**
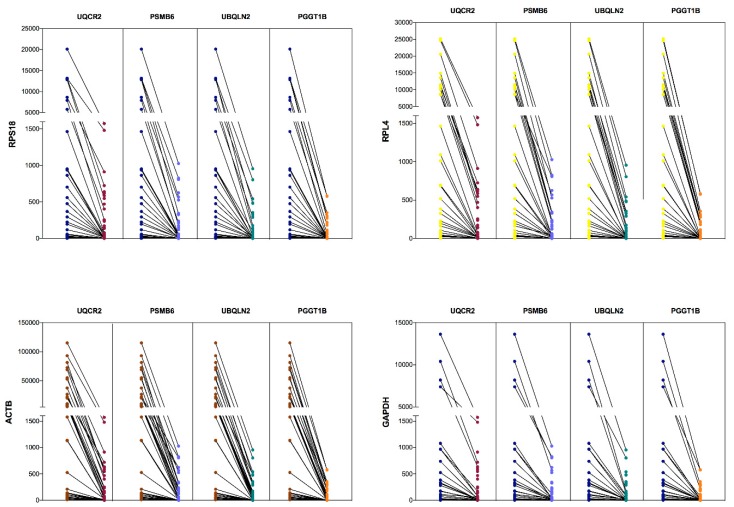
The gene expression comparison between the classical reference versus new reference genes in ALL. The top left plot shows the expression level of the *RPS18* gene versus *UQCR2*, *PSMB6*, *UBQLN2* and *PGGT1B*. The top right shows the level of *RPL4* gene expression versus *UQCR2*, *PSMB6*, *UBQLN2* and *PGGT1B*. The lower left shows the expression levels of *ACTB* versus *UQCR2*, *PSMB6*, *UBQLN2* and *PGGT1B*. The lower right shows the expression levels of the *GAPDH* gene versus *UQCR2*, *PSMB6*, *UBQLN2* and *PGGT1B*.

**Figure 6 genes-10-00376-f006:**
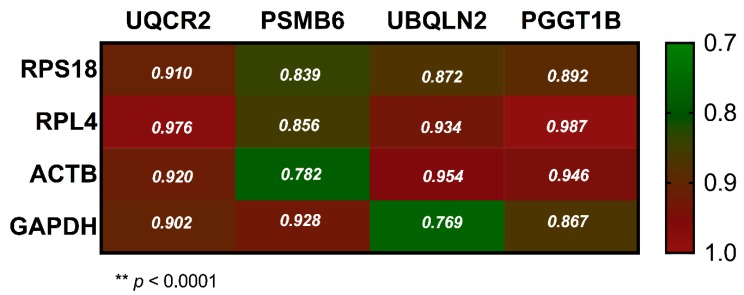
Pearson’s correlations of the classical references versus new reference genes. We observed significant correlations *p* < 0.0001 for all transcripts evaluated. The correlation is shown at the intersection. We observed that *UCQR2* and *PGGT1B* showed the best correlations with *RPS18*, *RPL4* and *GAPDH*. The lowest correlation was observed between *UBQLN2* versus *GAPDH* and *PSMB6* versus *ACTB* with 0.769 and 0.782, respectively.

**Table 1 genes-10-00376-t001:** Primer list.

Name	Sequences 5′ → 3′	Amplicon Size (bp)	Tm (°C)
*ACTB*	Fw 5′-TCACAATGTGGCCGAGGACTTT-3′ Rv 5′-AGAAGTGGGGTGGCTTTTAGGATG-3′	115	60
*GAPDH*	Fw 5′-CTCAACGACCACTTTGTCAAGCTC-3′ Rv 5′-CTCTCTTCCTCTTGTGCTCTTGCT-3′	147	60
*RPL4*-a	Fw 5′-CGAATGAGAGCTGGCAAAGGCAAA-3′ Rv 5′-ACGCCAAGTGCCGTACAATTCATC-3′	243	60
*RPL4*	Fw 5′- GTGGGACGTTTCTGCATTTG-3′ Rv 5′-TGTGCATGGGAAGATTGTAGT-3′	112	60
*RPS18*-a	Fw 5′-AATCCACGCCAGTACAAGATCCCA-3′ Rv 5′-TTTCTTCTTGGACACACCCACGGT-3′	241	58
*RPS18*	Fw 5′- CAGCCAGGTCCTAGCCAATG-3′ Rv 5′-CCATCTATGGGCCCGAATCT-3′	82	60
*UBQLN2*	Fw 5′-CAGCCTGAAGGATCAGTGTAGT-3′ Rv 5′-AGGGTCTCTTTATGGGAGAAGC-3′	84	60
*UQCR2*	Fw 5′-CCTGCGGGGTGATGTTGATA-3′ Rv 5′-CAGCTACTTCCCAACGACGA-3′	83	60
*PGGT1B*	Fw 5′-CTGTGGTTTCCGAGGCTCTT-3′ Rv 5′-GCATGAGAGGCCAGTGTAGG-3′	118	60
*PSMB6*	Fw 5′-TGACACCTATTCACGACCGC-3′ Rv 5′-GGACCAGTGGAGGCTCATTC-3′	129	60

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
