# Peer review of "Identification and Validation of Novel Reference Genes in Acute Lymphoblastic Leukemia for Droplet Digital PCR"

_genes, 2019, doi:10.3390/genes10050376_

Reviewer 1 Report

The manuscript entitled “Identification and validation of novel reference genes in acute lymphoblastic leukemia for droplet digital PCR” by Villegas-Ruiz et al. is an interesting paper that proposes four new reference genes (NRGs) for ddPCR gene expression assays. These NRGs may be very useful to normalize genes whose expression levels are very low, facilitating the comparison between samples and experimental conditions and increasing the accuracy of the results. However, some points should be addressed:

Please, consider revising lines 50-51. It seems as if one or more words were missing.

Lines 244-246: “Our results showed that 0.8 ng/well is enough to determine the expression of these transcripts (CRGs), while 20 ng/well is enough to evaluate the transcripts with low expression (NRGs)”. How many replicates of each sample were analyzed? In case only one replicate per sample has been analyzed, have the authors considered the possibility of combining several wells from the same sample to increase the sensitivity of detection in low expression genes?

Please consider the possibility of improving the resolution of Figure S2. Details of ddPCR analysis plots are hardly distinguishable. In Figure S2-A, ddPCR analysis plot from RPL4 shows that a different cut-off (pink line) has been applied to the NTC with respect to the other wells. Why? Could the observed events in the NTC well be due to contamination? Primer dimers? This point should be clarified.

Lines 334-335: “[…] the NRGs were expressed ~100 times more than the CRGs”. Please correct this sentence since actually the CRGs were expressed ~100 times more than the NRGs.

Did the authors analyze the expression levels of the 4 proposed NRGs in healthy controls by ddPCR? Were these genes expressed at low levels as reported in ALL patients? Were these genes analyzed in other childhood malignancies by ddPCR?

It would be recommended to describe in some detail the new reference genes proposed in this paper. Do the authors suggest any possible explanation for this 100-fold difference in expression between CRGs and NRGs? Is there any reason for these housekeeping genes to be under-expressed in childhood cancers? This point should be discussed.

Author Response

1.- Please, consider revising lines 50-51. It seems as if one or more words were missing.

Answer: The sentence was corrected 

2.- Lines 244-246: “Our results showed that 0.8 ng/well is enough to determine the expression of these transcripts (CRGs), while 20 ng/well is enough to evaluate the transcripts with low expression (NRGs)”. How many replicates of each sample were analyzed? In case only one replicate per sample has been analyzed, have the authors considered the possibility of combining several wells from the same sample to increase the sensitivity of detection in low expression genes?

AnswerAll samples were evaluated one replicate by ddPCR, because one of the advantages of this methodology is elimination of the technical replicas. In fact, target molecular are partitioning before individually amplification and each reaction is analyses separately; finally, the final reaction have an average of 15, 000-20,000 drops (Sanders, R., et al. PLOS One, 2013, 8(9):e75296). On the other hand, it is not possible combining samples, the mix reaction is performed in the PCR plate, and it is sealed to eliminate contamination. After, the sample is reading by droplet reader instrument. The ddPCR is a very sensitivity method, is possible to detected one copy of DNA/RNA (Taylor, S.C et al., Scientific Report, 2017, 7:2409, 1-8). 

3.- Please consider the possibility of improving the resolution of Figure S2. Details of ddPCR analysis plots are hardly distinguishable. In Figure S2-A, ddPCR analysis plot from RPL4 shows that a different cut-off (pink line) has been applied to the NTC with respect to the other wells. Why? Could the observed events in the NTC well be due to contamination? Primer dimers? This point should be clarified.

AnswerWe changed the figure by other the better resolution in Figure S2. In Fig S2-A, the cut off is different because we observed rain in the NTC, probably by dimer primers or self-complementary. In future assays we improve the assays, perhaps we changing the primers.

4.- Lines 334-335: “[…] the NRGs were expressed ~100 times more than the CRGs”. Please correct this sentence since actually the CRGs were expressed ~100 times more than the NRGs.

AnswerThe sentence has already been corrected.

5.- Did the authors analyze the expression levels of the 4 proposed NRGs in healthy controls by ddPCR? Were these genes expressed at low levels as reported in ALL patients? Were these genes analyzed in other childhood malignancies by ddPCR?

AnswerActually, the protocol is under review for approval by ethics committee for analyzed healthy controls, however, in the next study we will include controls and other childhood cancer. 

6.- It would be recommended to describe in some detail the new reference genes proposed in this paper. Do the authors suggest any possible explanation for this 100-fold difference in expression between CRGs and NRGs? Is there any reason for these housekeeping genes to be under-expressed in childhood cancers? This point should be discussed.

AnswerThe complete transcriptome is unknow in many tissues, we identify 98 transcripts that not had change their gene expression in cancer and healthy tissues, we do not have explanation because some genes are less expression than others. We think that the CRGs has been the most useful because their expression is the most evident in several tissues. However, in the ddPCR the high sensitivity and the absolute quantification, we can identify whit accuracy the number of transcripts expressed. For this, we need to found some new references genes that the expression is similar to problems genes.

Reviewer 2 Report

The paper by Villegas-Ruíz et al identifies four new reference genes for ddPCR. The identification of reference genes that do not show variations between patients and healthy conditions is a really important issue that must be considered when studying gene expression in cancer.

Major comments

Results, lines 203-205: After visual inspection, we selected ten transcripts (Figure 1, S1), but only four transcripts (PSMB6, PGGT1B, 204 UBQLN2 and UQCRC2) were validated (Figure 1). How did the authors validate those genes? Explain it in the text (not only in the figure legend).

Results, line 212: The authors say “To validate our results we selected four reported CRGs”. What did you want to validate?

Discussion, line 293: “RNA is a dynamic molecule whose expression is tissue-dependent; thus, it is very important to use the reference genes”. This sentence seems incomplete to me. Why is this important to this work? Moreover, the ideas of RNA being a “dynamic” molecule, with a changing expression, and the importance of using reference genes are discussed again later in the text  (lines 309-315).

Discussion, line 325: the authors say that “To date, there are no reference genes to use in ddPCR for GE”. Are they sure of this?

Minor comments

-       Lines 68-69: I don´t understand the sentence “It has higher accuracy and precision and advances”. In fact, the whole paragraph is confusing.

-       Line 77: what do the authors mean by “potential molecular GE”

-       Methods: why was the RNA quantified twice?

-       Table S1 in the text is named as “Table 1” in the supplementary data, while the table that appears as “Table 1” in the manuscript is in fact the primer list. I guess all the figures and tables contained in the document “Supplementary data” are supplementary. Thus, I would re-named them using the prefix “S”, for example: Table S1, instead of Table 1 again, which can be confusing. 

-       Line 296: “as”???

-       Table S2: Hodgkin lymphoma is repeated. The heading “number of microarrays” is not accurate, as they give the actual name of the microarrays, not numbers.

The manuscript has several gramatical and spelling errors, thus I think that it would really benefit from proofreading by a native speaker. For instance:

-       Line 51: the second “the GE process” seems to be repeated.

-       Line 52: It is not “bioinformatics studies”, but “bioinformatic studies”.

-       Line 80: instead of “problem gene”, it is more accurate “gene of interest”.

-       Lines 80-82: Does “diversity studies” intend to mean “several studies”?

-       Lines 83: lack of reproducibility.

-       Line 86: delete “the detection of”.

-       Line 90: successively level expression??

Author Response

Major comments 

- Results, lines 203-205: After visual inspection, we selected ten transcripts (Figure 1, S1), but only four transcripts (PSMB6, PGGT1B, 204 UBQLN2 and UQCRC2) were validated (Figure 1). How did the authors validate those genes? Explain it in the text (not only in the figure legend).Answer: Thank you for the observation, we validated those genes by quantitative RT-PCR and droplet digital PCR, the validation was included in the text line 219.

Results, line 212: The authors say “To validate our results we selected four reported CRGs”. What did you want to validate?Answer: The validate was changed by contrast, the term validate is not correct in the sentence, line 228.  

Discussion, line 293: “RNA is a dynamic molecule whose expression is tissue-dependent; thus, it is very important to use the reference genes”. This sentence seems incomplete to me. Why is this important to this work? Moreover, the ideas of RNA being a “dynamic” molecule, with a changing expression, and the importance of using reference genes are discussed again later in the text  (lines 309-315).Answer: Line 293, the sentence was deleted because it is discussed in the line 329. The complete idea is shown in the line 325-329.

Discussion, line 325: the authors say that “To date, there are no reference genes to use in ddPCR for GE”. Are they sure of this?Answer: The sentence was changed, To date, there are a few reference genes to use in ddPCR for GE, line 347.

Minor comments  Lines 68-69: I don´t understand the sentence “It has higher accuracy and precision and advances”. In fact, the whole paragraph is confusing.Answer:The sentence was modified, “It has higher accuracy, sensitivity and several advantages because it has minor inhibitory effects on PCR” line 69-70.

Line 77: what do the authors mean by “potential molecular GE” Sentence was modified “potential molecular GE markers” Methods: why was the RNA quantified twice?Answer: The Gene expression could be influenced by several technical factors, including contamination by reagent of RNA extraction, RNA integrity, RNA quantification by spectrophotometry and fluorometry and cDNA synthesis. In fact, we have both quantification methods and we wanted to see that quantification of total RNA not had drastic difference by both methods before making cDNA synthesis. However, our results not showed significant differences by quantification method.

Table S1 in the text is named as “Table 1” in the supplementary data, while the table that appears as “Table 1” in the manuscript is in fact the primer list. I guess all the figures and tables contained in the document “Supplementary data” are supplementary. Thus, I would re-named them using the prefix “S”, for example: Table S1, instead of Table 1 again, which can be confusing. Answer: The supplementary data were re-named.

Line 296: “as”???Answer: The sentence was modified, identify molecular patterns in several types of cancer, mutations..

Table S2: Hodgkin lymphoma is repeated. The heading “number of microarrays” is not accurate, as they give the actual name of the microarrays, not numbers.Answer: The Table S2 was corrected.

Line 51: the second “the GE process” seems to be repeated. -       Line 52: It is not “bioinformatics studies”, but “bioinformatic studies”. -       Line 80: instead of “problem gene”, it is more accurate “gene of interest”. -       Lines 80-82: Does “diversity studies” intend to mean “several studies”? -       Lines 83: lack of reproducibility. -       Line 86: delete “the detection of”. -       Line 90: successively level expression??Answer: The sentences was corrected